



# Regional analysis of parameter sensitivity for simulation of streamflow and hydrological fingerprints

Simon Höllering[1], Jan Wienhöfer[1], Jürgen Ihringer[1], Luis Samaniego[2], and Erwin Zehe[1]

[1]Karlsruhe Institute of Technology (KIT), Karlsruhe, Germany
[2]Helmholtz-Centre for Environmental Research, Leipzig, Germany

*Correspondence to:* Simon Höllering (simon.hoellering@kit.edu)

**Abstract.** Diagnostics of hydrological models is pivotal for a better understanding of catchment functioning. The analysis of dominating parameters for the simulation of streamflow plays a key role for region specific model diagnostics, model calibration or parameter transfer. A major challenge in this analysis of parameter sensitivity is the assessment of both temporal and spatial differences of parameter influences on simulated streamflow response. A methodical approach is presented, wherein a two-tiered global sensitivity analysis on a spatially distributed hydrological model is applied to 14 mesoscale headwater catchments of the river Ruhr in western Germany. The analysis of parameter sensitivity is geared towards two complementary forms of streamflow response targets. The analysis of the temporal dynamics of parameter sensitivity (TEDPAS) is contrasted with sensitivity analysis directed to hydrological fingerprints, i.e. temporally independent and temporally aggregated characteristics of streamflow (INDPAS). The two-tiered approach allows to discern a clarified sensitivity pattern pinpointed to diverse response characteristics, to detect regional differences and to reveal the regional relevance of the response target. Small local differences in the hydroclimatic and topographic setting of the headwaters lead to slight differences in the hydrological functioning, which was revealed by gradual differences in TEDPAS and INDPAS.

## 1 Introduction

### 1.1 Analysis of parameter influences

The role of hydrological model parameters has been studied for a long time. The illposed nature of hydrological modelling led to the awareness that parameter sets are not uniquely identifiable (Beven, 1993) and to the related branches of uncertainty assessment (e.g. Gupta et al., 1998) and automated parameter estimation (e.g. Hogue et al., 2000). This is closely related to the sensitivity of model results to parameter variations. While a number of topics are often subsumed under sensitivity analysis, underlying objectives and methodological approaches can substantially differ on a hydrological case-by-case basis (van Griensven et al., 2006; Saltelli et al., 2008; Zajac, 2010; Razavi and Gupta, 2015). Local and global strategies of sensitivity analysis have shown to be beneficial instruments at different stages of the modelling process (McCuen, 1973; Hamby, 1994; Sieber and Uhlenbrook, 2005; Razavi and Gupta, 2015). Analogous to the number of different objectives and methods to assess parameter sensitivity, results are subject to different forms of interpretation (Razavi and Gupta, 2015). The way the outcome of sensitivity analysis is evaluated and illustrated can strongly affect conclusions that are drawn. In this regard, results of





sensitivity analysis can widely differ if varying objective functions are considered for the evaluation of parameter influences (Demaria et al., 2007; Wagener et al., 2009); for a comprehensive overview see Reusser et al. (2011).

Time-integrated sensitivity measures (van Griensven et al., 2006; Sudheer et al., 2011; Nossent and Bauwens, 2012) alone merely allow more than rough estimates about the overall importance of parameters. Contrarily, McCuen (1973) pointed

out early that parameter sensitivity should be analysed in a time-dependent context, as hydrological systems are subject to temporally dynamic processes. Guse et al. (2016) argued that the study of temporal variations in sensitivity is essential to learn about the relation between dominant parameters and governing processes under changing hydrological conditions to be reflected in the model results. The characterisation of temporal dynamics of parameter sensitivity (TEDPAS) has been accomplished in diverse ways (Cloke et al., 2008; Cibin et al., 2010; Reusser et al., 2011; Reusser and Zehe, 2011; Herman

et al., 2013; Sanadhya et al., 2013; Guse et al., 2014; Pfannerstill et al., 2015; Pianosi and Wagener, 2016). The choice of the temporal resolution is an important factor which clearly influences the way parameters are identified and how inferences on related processes are made (Tang et al., 2007; Massmann and Holzmann, 2012; O'Loughlin et al., 2013). Necessarily, the timescale of sensitivity analysis is selected in accordance with the objective of the study and the dynamics of the system under investigation. The importance of parameters temporally varies in a similar way like short periods of high flows alternate with

longer periods of low flow (Massmann et al., 2014).

When model calibration and verification comes into play, analysis of parameter sensitivity provides valuable information on the importance of each input factor in regard to simulated model output. On this basis, it can be decided for each parameter if its value should be determined exactly, or if it could even be completely excluded, fixed at predetermined values (Reusser et al., 2011). Preferably, sensitivity analysis minimises the necessary number of parameters as hydrological models are often

subject to overparameterisation (Beven, 2001; Kirchner, 2006; van Werkhoven et al., 2009; Samaniego et al., 2010b).

A common goal of sensitivity-guided studies dealing with an identification of dominant processes is the achievement of a suitable representation of real-world hydrological processes through understanding the reasons for model defectiveness. If non-sensitive parameters are detected, an indication of model structural deficits (Kirchner, 2006; Gupta et al., 2012), or a lack of the adequate model response target data might be given. Sensitivity analysis has not just recently deemed helpful as a

diagnostic tool to identify structural and performance deficits of hydrological models (McCuen, 1973; Sieber and Uhlenbrook, 2005; Yilmaz et al., 2008; Kavetski and Clark, 2010; Guse et al., 2014; Pfannerstill et al., 2015). Reusser and Zehe (2011) showed that a combined analysis of the temporally varying parameter dominance (sensitivity analysis) and model performance (error analysis) can be applied to effectively detect structural inadequacies of model components for a specific landscape.

## 1.2 Fingerprint based sensitivity analysis

The characterisation of hydrological process complexity as the functioning of catchments can be addressed in various ways, at multiple scales and levels of complexity. Fingerprint metrics (hereinafter also referred to as fingerprints) are signatures of dynamic catchment response that change on different temporal and spatial scales (Sivapalan, 2005; Wagener et al., 2007; Winsemius et al., 2009).



In hydrological modelling, multiple fingerprint metrics have been adopted to enhance model evaluation beyond the minimisation of streamflow residuals. Fingerprints of catchment functioning may be technically distinguished into measures based on single (statistical) streamflow indices and on characteristic curves, e.g. (cumulative) frequency curves, regime curves, or double mass curves. Single representatives of both categories can be selected to describe single components of streamflow regimes, namely the magnitude, frequency of occurence, duration, timing and flashiness of flow events (Poff et al., 1997; Olden and Poff, 2003), or of the general hydrological variability at different spatial and temporal scales.

In a comprehensive analysis of catchment functioning in order to understand dominant processes, the use of single criteria is not sufficient. Multi-variable approaches based on hydrological fingerprints have been the method of choice. Fingerprints have been jointly used as multivariate objectives to estimate the parameters of hydrological models (Shamir et al., 2005a, b; Pokhrel et al., 2008; Castiglioni et al., 2010) or to assess model performance and evaluate model structures (Farmer et al., 2003; Gupta et al., 2008; Yilmaz et al., 2008; Clark et al., 2011; Euser et al., 2013; Vrugt and Sadegh, 2013).

Sensitivity analysis related to streamflow characteristics has formerly mostly been applied prior to model evaluation (e.g. Atkinson et al., 2003). More recently, with the aid of observed quantities, sensitivity analysis was also directed to fingerprint performance measures as objective functions (e.g. van Werkhoven et al., 2008; Yilmaz et al., 2008; van Werkhoven et al., 2009; Sanadhya et al., 2013) or to the analysis of temporally resolved process sequences (e.g. Pfannerstill et al., 2015). Other studies applied fingerprint metrics but based their analysis of parameter sensitivity only on few aspects of streamflow (e.g. limb densities; Shamir et al., 2005a) or on single (statistical) streamflow indices of different aggregation timescales (Shamir et al., 2005b).

In our view, sensitivity analysis geared towards fingerprint metrics as multivariate response target has not received adequate consideration for model diagnostics. Especially in terms of joint fingerprints, using both single value indices and characteristic curves along independent variables, the full potential for process-oriented model diagnostics has not been exploited. Some progress has been made from the other side by Guse et al. (2016) who combined TEDPAS for different temporal resolutions with segments of the flow duration curve (FDC) to identify parameters and related processes that dominate at variable streamflow magnitudes of two distinct streamflow regimes.

## 1.3 Objectives, research questions and approach

The main objectives of this study are to analyse the parameter sensitivity of a mesoscale hydrological model for the simulation of streamflow response and hydrological fingerprints at a set of headwater catchments of the Ruhr in Germany. The approach extends the temporally dependent analysis of parameter sensitivity (TEDPAS) along two avenues: The first is to investigate TEDPAS results in more detail to derive parameter sensitivities in different hydrological conditions; the second is to direct the analysis to other, temporally independent characteristics of streamflow response (INDPAS).

With this approach we explore the following three research questions:

– Which parameters can be identified as sensitive with regard to specific hydrological response characteristics?





- How does parameter sensitivity change with different hydrological objectives (response targets) applied in global sensitivity analysis?

- How does parameter sensitivity change in different catchments under slightly distinct physiographic and hydroclimatic conditions?

The methodological approach rests on the combination of first-order partial parameter sensitivity of a state-of-the-art distributed hydrological model to simulated streamflow hydrographs and the related temporally independent and time-aggregated streamflow response characteristics. The analysis is structured in the following steps:

- Combining the application of a hydrological model as a learning tool and forward operator with global sensitivity analysis;

- Deriving fingerprint metrics (single value indices and characteristic curves) from simulated streamflow time series;

- Analysing parameter sensitivity to temporally resolved dynamics of streamflow response (TEDPAS);

- Analysing parameter sensitivity both to temporally aggregated (single value indices) and to temporally independent (characteristic curves) characteristics of streamflow (INDPAS);

- Assessing differences in parameter sensitivity between the two different methodical approaches (TEDPAS and INDPAS),
and between the analysed headwaters.

In the study we will thus complement sensitivity analysis based on temporally dependent output variables (TEDPAS) with both temporally independent (characteristic curves) and temporally aggregated single-valued (fingerprint metrics) characteristics of streamflow response (INDPAS). In cases where characteristic curves (e.g. the FDC) are used as fingerprint metrics, changes in parameter sensitivity will be analysed for changes in the independent variable (e.g. streamflow exceedance prob-
ability). We focus the study on the headwaters of the Ruhr catchment (section 2.6.1) in western Germany also for regional analyses, based on available data sets (section 2.6.2).

From this we expect to pinpoint dominant parameters related to individual process components and to ease the interpretation of parameter sensitivity detached from the variability of timescales. Bearing in mind the complexity of the evaluation of spatially and temporally distributed model responses, our multilateral approach aims at providing further insight into the
dominance of model parameters and related streamflow response processes.

## 2  Methods and models

First we detail the fingerprint metrics used to characterise streamflow response (section 2.1). We implemented the Fourier Amplitude Sensitivity Test (FAST; section 2.2) to conduct sensitivity analysis of the mesoscale Hydrologic Model (mHM; section 2.3) in the Ruhr headwater catchments with a focus on eight global mHM parameters (section 2.4) basically employing two





different forms (TEDPAS and INDPAS) of simulated streamflow response (section 2.5). Finally, we introduce the catchment of the river Ruhr and the headwaters which were selected for this study (section 2.6.1), and specify the data used to conduct the analysis (section 2.6.2).

## 2.1 Fingerprint metrics

The fingerprint metrics used for characterising the hydrological response of the investigated catchments included single value indices and the flow duration curve as an example for catchment characteristic curves. These fingerprints were derived from model results and precipitation data (see section 2.6.2), respectively.

We chose eight indices reflecting different aspects of the integral and longterm hydrological functioning of catchments in a single, time-aggregated number (Table 1). These fingerprints characterise the overall water balance (Runoff Ratio, RR), the
variability of streamflow (Coefficient of Variation, CV), the frequency of flow events (High Pulse Count, HPC), the change rate of streamflow (Slope of Flow Duration Curve between 33 % and 66 %, SLFDC), the streamflow during high flow (High Flow Discharge, HFD) and low flow (Baseflow Index, BFI) conditions, the streamflow recession behaviour (Recession Time Constant, RTC), and the autocorrelation structure of streamflow (Autocorrelation Time, ACT), respectively. In this study, the slope of the flow duration curve (SLFDC) is the only single value fingerprint that could not be directly determined from
streamflow hydrographs. Instead, the FDC was used as a basis for its derivation. The eight single value fingerprints were implemented as model response targets for sensitivity analysis (section 2.5.2).

As an example for more complex characteristics than single-valued indices, we also used entire flow duration curves as a model response target for sensitivity analysis (section 2.5.2).

## 2.2 Fourier Amplitude Sensitivity Test (FAST)

FAST is a partial variance-based method to determine global sensitivities of parameter changes on the outcome of monotonic and non-monotonic numeric models (Cukier et al., 1973; Schaibly and Shuler, 1973; Cukier et al., 1975). The general idea of FAST is a) to vary parameters of interest along a predefined number of model runs with independent frequencies, and b) to perform a Fourier analysis at each time step across the ensemble of model runs. The amplitudes of the different frequencies reveal the first-order sensitivities of the related parameters. For a detailed explanation of the FAST method the reader is referred
to Reusser et al. (2011).

FAST was originally applied to study parametric model sensitivities of chemical reaction systems. In recent decades, the method has been used and evaluated in a variety of fields such as hydrogeology (Fontaine et al., 1992), atmospheric sciences (Rodríguez-Camino and Avissar, 1998), geologic nuclear waste disposal modelling (Lu and Mohanty, 2001), food-safety risk assessment (Frey and Patil, 2002), or ecologic forestry (Song et al., 2013). A number of studies treat the application of FAST
in hydrological modelling (van Griensven et al., 2006; Zajac, 2010; Reusser and Zehe, 2011; Sanadhya et al., 2013; Guse et al., 2014; Pfannerstill et al., 2015; Guse et al., 2016).

FAST is a highly efficient computational method that requires significantly fewer model runs to yield similar results for parameter sensitivity in comparison with other approaches (Saltelli and Bolado, 1998; Reusser et al., 2011). The number of





model runs (hence parameter sets) in FAST is determined by the number of analysed model parameters. That means always the same number of model runs is required for a given number of parameters, independent of model, catchment or (type of) parameter.

## 2.3 mesoscale Hydrologic Model (mHM)

The mesoscale Hydrologic Model (mHM; Kumar et al., 2010; Samaniego et al., 2010b) accounts for diverse processes of the hydrological cycle: Canopy interception, evapotranspiration, snow, soil moisture dynamics, overland flow, infiltration, interflow, subsurface storage, groundwater recharge, baseflow, discharge attenuation as well as flood routing. The mHM is conceptualised on the basis of grid cells, and has been applied to a wide range of mesoscale river catchments ($10^1 - 10^4 \, \mathrm{km}^2$; Kumar et al., 2010; Samaniego et al., 2010a, b, 2011; Cuntz et al., 2015; Rakovec et al., 2016a, b). Gridded information

is implemented in mHM at three levels: morphology (level 0), hydrology (level 1), meteorology (level 2), with $l_0 \ll l_1 \le l_2$ denoting the relative sizes of the grid cells at the respective data level (Kumar et al., 2010).

The parameterisation of mHM is substantially based upon a simultaneous regionalisation technique called multiscale parameter regionalisation to account for the physiographic sub-grid and hydrological process variability (Samaniego et al., 2010b; Kumar et al., 2013). Hydrological process parameters at level 1 are derived from physiographic characteristics at level 0 using

(pedo-)transfer functions with predefined coefficients (in the following referred to as global mHM parameters). Hence, mHM is calibrated indirectly, by altering the 52 level-0 parameters of the transfer functions instead of the hydrological level-1 parameters. This procedure not only reduces the problem of overparameterisation or the dependence on specific hydrological scales (Beven, 2001) but also reduces the amount of time to be spent on grid-wise calibration (Samaniego et al., 2010b).

## 2.4 Model setup for sensitivity analysis

In this study both the hydrological model level $l_1$ and the meteorological $l_2$ of mHM were set to a spatial resolution of 1 km, whereas for level 0 with the physiographic catchment data (morphology), a finer resolution of $l_0 = 200$ m was selected as an adequate spatial discretisation. Model simulations were conducted for each of the 14 headwater catchments (see Fig. 1) with a daily timestep for the ten years period of 1997 to 2006.

To facilitate the selection of the most sensitive parameters, we first carried out a preliminary local FAST analysis at gauge

Wenholthausen (WEN; see Fig. 1) including all 52 global mHM parameters to reveal parameter sensitivities to streamflow simulations. For this initial analysis, 21803 model runs were conducted and the streamflow hydrographs were analysed with FAST. We found 14 parameters with a maximal sensitivity value of more than 0.01 (1 %). By inspection of the model equations we identified correlations between these parameters, which led to the removal of six parameters from this set.

The eight uncorrelated parameters (Table 2) were used for the regional sensitivity analysis in the 14 headwater catchments.

All other mHM parameters were kept fixed on calibrated values found via global automatic optimisation using the dynamically dimensioned search algorithm (Tolson and Shoemaker, 2007) at WEN. The value ranges for the parameters were basically selected from mHM literature (Samaniego et al., 2014), partly extended based on the results from the preliminary analysis. For eight parameters, the FAST method requires 243 model runs based on different combinations originating from FAST





parameter variation with independent frequencies inside the parameter ranges (see Fig. 2). The same 243 combinations of mHM parameter sets were used for streamflow simulations in each of the 14 catchments. Differences between catchments in terms of hydroclimatic forcing and physiographic attributes were included in the model by the locally specific meteorological and morphological input on data levels $l_2$ and $l_0$.

## 2.5 Sensitivity analysis

We analysed the parameter sensitivity in different forms to be able to evaluate the dominance of parameters and to potentially detect local differences among the headwaters related to various aspects of streamflow response functioning in a more specific way. We first used simulated streamflow hydrographs (TEDPAS; section 2.5.1) and then both temporally aggregated (fingerprint metrics) and temporally independent (FDCs) characteristics of simulated streamflow response (INDPAS; section 2.5.2) as model response targets for the sensitivity analyses.

### 2.5.1 TEDPAS - Temporal dynamics and sensitivity duration

Using simulated hydrographs with FAST provided daily time series of partial parameter sensitivities for each headwater catchment for the simulation period 1997-2006. These temporal dynamics of parameter sensitivity (TEDPAS; Reusser et al., 2011) were analysed and compared for the Ruhr headwater catchments (section 3.1).

We also calculated Sensitivity Duration Curves (SDCs) for each parameter, which we defined in analogy to other well-known cumulative frequency curves like the FDC. Each SDC is specific for one of the eight parameters, for one (type of) catchment and for the period (1997-2006) in which sensitivity analysis is performed. SDCs were developed for each catchment by arranging the daily sensitivity values from FAST by magnitude in ascending order and by plotting them as a line against the percentage of time during which the sensitivity equalled or exceeded the specified values. Sensitivities were normalised by the highest sensitivity value found for each parameter among all headwaters. These curves reveal whether a parameter is consistently (non-)sensitive or if its importance changes during the simulation period (section 3.2).

### 2.5.2 INDPAS - Parameter sensitivity to fingerprint metrics

For each catchment we calculated the eight single-valued fingerprint metrics (section 2.1 and Table 1) from each of the 243 simulated streamflow hydrographs. By applying FAST we obtained the partial sensitivities of the model parameters with regard to each fingerprint (section 3.3.1).

In a similar way, for each headwater catchment, 243 flow duration curves were derived from the simulated streamflow time series and analysed with FAST. This yielded parameter sensitivities along the axis of streamflow exceedance probability as an independent variable, revealing which parameters dominate streamflow simulations during high, intermediate or low flow conditions. As a supplementary step, the parameters showing the highest sensitivity for a given streamflow exceedance probability were extracted along the independent variable, revealing patterns of dominant parameters over the spectrum of streamflow in each headwater catchment (section 3.3.2).



## 2.6 Study area and data

### 2.6.1 The Ruhr headwater catchments

The Ruhr has a catchment area of 4.485 $km^2$, and originates from a spring at about 670 m a.s.l. on the northern slope of the Ruhrkopf (842 m a.s.l.). The Ruhr joins the Rhine at Duisburg-Ruhrort (20 m a.s.l.) after 219 km (Fig. 1). The landscape characteristics of the catchment range from densely wooded and scarcely populated lower mountain ranges in the Sauerland to widely sealed urban areas in the river valleys and in the western part close to the mouth. The area belongs to the geology and geography of the Rhenish Slate Mountains to the east of the Rhine (Brudy-Zippelius, 2003). The average discharge at the confluence with the Rhine is about 80.5 $m^3 s^{-1}$ (Bode et al., 2003). With a total of eight dams and five reservoirs the Ruhr and its tributaries form a complex hydrological system. The total stored water surface area of about 35 $km^2$ equates to about 480 million $m^3$ of water retained behind damming structures (Ruhrverband, 2011). An intensive use of water resources (e.g. reservoirs, barrages, withdrawals, inlets, etc.) supplies almost 5 million people with drinking and processing water along the Ruhr, within its catchment and to adjacent watersheds.

Our investigations concentrate on 14 headwater catchments of the river Ruhr and its tributaries (e.g. Bigge, Lenne and Möhne), where the hydrological regimes are much less affected by water management measures (Fig. 1). The headwaters are situated in the eastern, rural part of the Ruhr basin with higher altitudes, and cover an area of 1989.2 $km^2$ in total. Individual catchment sizes range from 28.7 $km^2$ at gauge Amecke (AME) to 453.1 $km^2$ at gauge Bamenohl (BAM). Average catchment slopes vary between 10.8 % (Rüblinghausen, RUE) and 26.1 % (Kickenbach, KIC). The dominant form of land cover is forest (39.7 % - 87.3 %) followed by pasture (0.8 % - 47.5 %), cropland (7.6 % - 43.9 %) plus a few, predominantly dispersed, settlements (0.0 % - 13.2 %; Table 3). The climatic conditions are humid-warm-temperate (Göppert et al., 1998) with warm summers and moderate winters. Annual mean temperature ranges between 8.45 °C and 5.45 °C at the lower and higher altitudes in the study area, respectively. Annual precipitation ranges from 1025 mm in the northeast to 1425 mm in the southwest (1997-2006; Table 3).

### 2.6.2 Data

Different kinds of observation data were used to set up and calibrate the hydrological model, to perform simulations for sensitivity analysis, to derive the fingerprint metrics and a set of physiographic catchment descriptors.

Meteorological input data were daily values for precipitation (HYRAS; Rauthe et al., 2013), temperature (HYRAS; Frick et al., 2014), and potential evapotranspiration (AMBAV; Löpmeier, 1994), all at a spatial resolution of 1 $km^2$. For the calibration of mHM at gauge Wenholthausen observed streamflow data from 2002 to 2006 were available. Spatial physiographic data were a digital elevation model (50 m x 50 m), CORINE land cover data (100 m x 100 m; European Environment Agency, 2009), a soil map (1:200.000; Bundesanstalt für Geowissenschaften und Rohstoffe, 2015a) and a geological map (1:1.000.000; Bundesanstalt für Geowissenschaften und Rohstoffe, 2015b).

A set of 14 descriptors to characterise the hydroclimatic and physiographic setting of the headwaters and to capture characteristics that might jointly control relevant hydrological functions as defined by Black (1997) has been compiled in Table 3.





Each descriptor in Table 3 was assigned to one of five main classes of catchment characteristics, i.e. climate (1), landform (2), topography (3), land cover (4) and soil (5), as proposed by Yadav et al. (2007). The choice of climate and physiographic descriptors originates from correlation analysis of catchment descriptors within each category (Yadav et al., 2007), multivariate statistical analysis techniques (Di Prinzio et al., 2011), regionalisation models (Plate et al., 1988), GIS based analysis of the

digital elevation model and, in the case of the baseflow index (BFI), from comparison of methods for baseflow separation (Duband et al., 1993). Table 3 includes the BFI as an intermediate form between physiographic descriptor of soil hydrological characteristics and temporally aggregated fingerprint metric introduced in section 2.1.

## 3  Results

### 3.1  Temporal dynamics of parameter sensitivity (TEDPAS)

TEDPAS analysis for the 14 headwaters in the period of 1997-2006 showed a strong temporal dependency of the fraction of the total variance explainable by first-order sensitivities for hydrograph simulation. The sum of all eight parameter sensitivity values per time step ranged between 0.26 and 0.87, while the average sum of the eight sensitivity values per time step was 0.71. The spread between the maximal and minimal sum per time step was found to be smaller in the southwestern (e.g. Rüblinghausen RUE, 0.48) than in the northeastern (e.g. Völlinghausen VOE, 0.61) headwaters.

Minimal and maximal (Sensitivity Range) and the average (Sensitivity Mean) sensitivity values of the eight parameters, summarised across all headwaters (Table 2), give a first impression that the soil moisture parameter Ksconst generally exhibited the highest influence (Sensitivity Mean 0.392), while AspectcorrPET showed the largest range (Sensitivity Range 0 - 0.78). The interflow parameter ExpslowInterflow had the smallest sensitivity range (0 - 0.22), whereas parameters for snow (DegdayForest) and baseflow (GeoParam) had the overall lowest sensitivity mean values of 0.012.

TEDPAS did not reveal many differences between the headwaters. For instance, Ksconst consistently had a highly dynamic course of sensitivity with frequently high values (Fig. 3a and b). Nevertheless, some of the parameters showed differences between the headwaters, for example for DegdayForest (January - March; Fig. 3a and b) and AspectcorrPET (November - April; Fig. 3c and d). AspectcorrPET allows to include the exposition of slopes, controlling insulation, in evapotranspiration estimations, while DegdayForest is a parameter related to snow dynamics in forested areas.

The example of these two parameters also illustrates the seasonality in sensitivity dynamics. AspectcorrPET showed highest sensitivity in the summer period from April to August, when evapotranspiration processes dominate and streamflow dynamics are low (Fig. 3c - f). During that period, the parameter showed an alternating course of sensitivity compared to Ksconst (Fig. 3b and d) with local maxima connected to (simulated) streamflow peaks (Fig. 3f).

Higher sensitivities of DegdayForest were found for periods (e.g. February) when snow processes (accumulation and melt-
ing) can occur. This was predominantly observed in catchments at higher altitudes, for example, rather in the headwater of VOE (up to 630 m a.s.l.) than in the one of RUE (450 m a.s.l; Fig. 3a and b). Additionally, VOE (50 %) exhibits a slightly higher percentage of forest cover (FOR) than RUE (43 %; Table 3). A similar distinction between summer and winter patterns was found for InfilShapeFactor, although at lower sensitivity levels (Fig. 3a and b). For the rest of the parameters either no





seasonal patterns (e.g. RechargeCoeff; Fig. 3c and d) could be revealed, or only very low sensitivity values were found (e.g. ThetaSconst; Fig. 3a and b).

The ensembles of simulated streamflow compared reasonably well with the observed hydrographs (Figure 3e and f), although the simulation ensemble underestimated some high flow periods.

## 3.2 Sensitivity duration

Sensitivity duration curves for the 14 headwaters revealed distinct influences of the eight parameters on streamflow simulations (Fig. 4). Different sensitivity characteristics were identifiable among the parameters, with either very low (e.g. DegdayForest; Fig. 4a and ThetaSconst; Fig. 4b), intermediate (RechargeCoeff; Fig. 4g) or high (Ksconst; Fig. 4c) influence along the independent axis of sensitivity exceedance probability.

Some of the parameters showed a regional variation of SDCs. Four of eight parameters, i.e. Ksconst (Fig. 4c), InfilShapeFactor (Fig. 4d), AspectcorrPET (Fig. 4e) and ExpslowInterflow (Fig. 4f), revealed certain differences among the headwaters. The SDCs of the two most influential parameters Ksconst (Fig. 4c) and AspectcorrPET (Fig. 4e) showed a systematic spread for the different headwaters, with the curve of gauge RUE plotting at the lower (Fig. 4c) and upper (Fig. 4e) margins of the group of headwaters, respectively. For InfilShapeFactor (Fig. 4d) the headwater of gauge Möhnesee-Neuhaus (MOE) deviated in direction of lower values, while for ExpslowInterflow (Fig. 4f) again both RUE and MOE showed divergent, low SDCs in the set of 14 headwaters.

In the case of AspectcorrPET (Fig. 4e) the SDCs were sorted from the southwestern (e.g. RUE) to the northeastern (e.g. MES) headwaters (Fig. 1). In the southwestern headwaters (e.g. RUE) the slopes are more gentle with lower relief energy than further northeast, where valleys are more deeply incised (e.g. NIC, SLOPE and ELR; Table 3). The slopes in the southwestern headwaters are on average directed to the southeast, compared to the more southwest-directed slopes in the northern and eastern Ruhr headwaters (EXP; Table 3). Besides showing a different aspect, the southwestern headwater RUE also has the highest proportion of urban areas (13 %; Table 3). Both factors influence the estimation of evapotranspiration in mHM and hence streamflow simulations.

The SDCs of the most sensitive parameter Ksconst (Fig. 4c) showed concave curvatures, in contrast to the other parameters which had convex SDCs. Except for ExpslowInterflow (Fig. 4f), the SDCs of the largest headwater Bamenohl (BAM) fell in between the other catchments, showing a kind of transitional behaviour of sensitivity duration (Fig. 4).

## 3.3 Parameter sensitivity to temporally independent fingerprints (INDPAS)

### 3.3.1 Single value indices

Similar patterns of parameter sensitivities to single value fingerprints were consistently found across all 14 headwaters. Figure 5 shows the matrix representations for four representative headwaters (RUE, VOE, HER and WEN). All of them comprise eight rows for the parameters and eight columns for the fingerprint metrics. Sensitivity to a specific fingerprint is arranged column-wise.



As in the TEDPAS analysis, Ksconst was the by far most sensitive parameter for the simulation of five of the fingerprints (CV, HPC, HFD, BFI, RTC) in all 14 headwater catchments. The parameters ExpslowInterflow and RechargeCoeff were identified as the second and the third most sensitive parameter in these cases. In terms of the fingerprint Runoff Ratio (RR), in contrast, AspectcorrPET was the most sensitive parameter, while others, including Ksconst, showed almost no sensitivity

to the simulation of the overall water balance. The parameters RechargeCoeff and Ksconst were of similar importance for the simulation of the fingerprint SLFDC (slope of the flow duration curve). Other parameter-fingerprint combinations revealed parameters with very low sensitivity values. Very low sensitivity, e.g. for DegdayForest or GeoParam were found with all of the eight fingerprint metrics (Fig. 5).

Only minor differences in these patterns occured between the catchments, and these related to small deviations in absolute
sensitivity values or in the order of the second and third rank, e.g. for the fingerprint ACT (Fig. 5).

### 3.3.2   Flow duration curve

Using FDCs as model response targets revealed parameter sensitivities to different streamflow magnitudes. Again, a high portion of similarities among the headwaters was found. The highest influence was alternately exerted by the parameters Ksconst and AspectcorrPET (Fig. 6a-d); their courses of parameter sensitivity were highly anticorrelated (mean correlation
over all headwaters $r = -0.975$). The soil moisture parameter Ksconst clearly dominated the very high flows (0 - 10 % of time Q is exceeded) and the entire mid and low flow sections (40 - 100 %); moderate high flows between 10 % and 40 % were most affected by changes in the evapotranspiration parameter AspectcorrPET. These changes in the dominating parameter are additionally illustrated in Fig. 6 by a catchment-specific strip showing the pattern of parametric dominance along the FDC, which showed only slight differences between the headwaters in the length of the intermittent parts (AspectcorrPET).

The other parameters reached overall lower sensitivity levels. The patterns were again similar for all headwaters, with minor differences regarding the absolute sensitivity values and the order of importance in the third and higher ranks. The parameters RechargeCoeff and ExpslowInterflow revealed a bimodal sensitivity distribution. RechargeCoeff showed a first peak between 0 and 20 %, and a steady increase from 40 % to its maximum sensitivity value at very low flows with 100 % of streamflow exceedance, which was a sensitivity value of about 0.25 in the case of gauge HER (Fig. 6c). ExpslowInterflow had its highest
parametric influence at very high flows (0 - 15 %), and at moderate to low flow magnitudes. The curves of InfilShapeFactor alternated with ExpslowInterflow along the FDCs (Fig. 6a-d), while the rest of the parameters did not show notable sensitivity values.

Interestingly, the ensembles of normalised FDCs showed distinct differences between the catchments (Fig. 6e-h), although the sensitivity dynamics were similar, and the same 243 parameter variations from FAST were used for each headwater. The
largest spread of FDCs was found for the northeastern headwater VOE (Fig. 6f) with the largest catchment size among the four shown headwaters. The smaller headwaters WEN, HER and RUE (Fig. 6h, g, e) showed a decreasing spread of the FDCs from northeast to southwest (Fig. 1). Additionally, the widths of the simulation envelopes were changing for different streamflow magnitudes. The ensembles of FDCs all showed a constriction point located at about 20 % of streamflow exceedance (Fig. 6e-h), which is the same point where Ksconst and AspectcorrPET showed lowest and highest sensitivity values, respectively





(Fig. 6a-d). The ensembles of simulated FDCs encompassed the observed FDC in most cases (e.g. VOE, HER, WEN; Fig. 6f-h). In some cases the observed FDC was outside the simulated range (e.g. RUE; Fig. 6e).

# 4   Discussion

## 4.1   Parameter sensitivities from TEDPAS and INDPAS

Sensitivity analysis for eight mHM parameters with TEDPAS and INDPAS in 14 Ruhr headwaters revealed that soil moisture and evapotranspiration parameters Ksconst and AspectcorrPET are most relevant for the simulation of the hydrological streamflow response. TEDPAS analysis provided information about seasonalities in sensitivity dynamics of parameters, controlling snow (DegdayForest) or evapotranspiration (AspectcorrPET) processes, in relation to simulated streamflow hydrographs. The results of TEDPAS were further used to analyse sensitivity duration. The form of the resulting SDCs differed according to the

importance of each parameter; furthermore, the SDCs allowed to detect regional differences (section 4.2). INDPAS analysis showed that the influential model parameters show changing sensitivities to different characteristics and to varying magnitudes of streamflow.

For very long periods of time, a temporally resolved sensitivity (daily streamflow time series) makes it difficult to reveal clear patterns of dominant parameters. Guse et al. (2016) also recognised that parameter sensitivity by TEDPAS based on

the streamflow hydrograph should be analysed on different temporal aggregation levels and related to different streamflow magnitudes (FDCs). These steps were found to be beneficial in order to reach clarified patterns of temporal process dynamics for a detailed assessment of dominant model parameters. While their methodological approach was purely based on aggregation and reordering of TEDPAS sensitivity and streamflow time series, we added additional value with INDPAS aiming at multiple response targets.

The application of single value fingerprints as model response targets for sensitivity analysis helped to elucidate further details of parametric dominance. For instance, clearly highest relevance for the RR can be attributed to AspectcorrPET in all of the headwaters (Fig. 5), which all are characterised by a high percentage of forest cover (Table 3). In a qualitative sense and in this specific case, exposition together with the vast forest coverage (transpiration) seems to play a major role for the overall water balance characterised by the runoff ratio. The high sensitivities of AspectcorrPET directed to RR indicate that the water

balance as one specific characteristic of the regional hydrological functioning is, in the model domain, predominantly controlled by AspectcorrPET. Thus, dominant parameters can be additionally adjusted pinpointed to fingerprint metrics in a subsequent calibration step. Likewise, a missing sensitivity signal to a fingerprint in the form of a single value index (e.g. RechargeCoeff to RR) can reveal that the chosen response target might not be relevant to further constrain parameter identification in a certain catchment. In the INDPAS analysis to single value fingerprints we identified different cases where parameters did not show

a clear sensitivity signal. These parameters and the fingerprint metric can be collectively removed from further consideration, e.g. for model calibration directed to the respective response target (van Werkhoven et al., 2009).

The spread in simulated values for the Autocorrelation Time (ACT) was moderate. Accordingly, INDPAS analysis directed to ACT also showed moderate sensitivities for the set of eight parameters (Fig. 5). Our selection of eight parameters was based





on the most sensitive ones with regard to the simulation of streamflow hydrographs (TEDPAS). This preselecting step can potentially lead to the elimination of other storage parameters that might be most sensitive to ACT. Thus, one might conclude that parameter selection based on INDPAS would result in a different choice in the set of the most sensitive parameters. But, in the case of the Autocorrelation Time this is not very likely, since storage parameters (e.g. ExpslowInterflow and GeoParam)

were still included in the set of eight parameters.

Our different findings when analysing different model response targets confirm the necessity of a multivariate sensitivity analysis. This was similarly recognised by Wagener et al. (2009) who applied three standard error metrics, e.g. the Root Mean Squared Error (RMSE), as objective functions for sensitivity analysis. Their results for parameter sensitivity were found to change spatially when the objective function was replaced. To avoid misinterpretation of sensitivity results we propose that the

selection of fingerprint metrics as model response targets should be determined by the purpose of the analysis. For instance, in studies where the goal is the prediction of floods, the focus must be on parameters showing sensitivity to fingerprint metrics for peak flow. Razavi and Gupta (2015) similarly pointed out that even conflicting conclusions could be drawn if different properties of the model response are applied in sensitivity analysis.

The consideration of flow duration curves enabled analysing streamflow free of autocorrelation and time dependency. The

FDC as model response target for sensitivity analysis provided information on parameter sensitivity along the independent variable of streamflow exceedance probability. In contrast, for classical hydrograph inspection, which is the basis of TEDPAS, time is the independent variable. INDPAS along FDCs allowed to draw conclusions about parametric influences at specific streamflow magnitudes.

In our case, Ksconst and AspectcorrPET as conceptual proxies for soil moisture dynamics and evapotranspiration processes

alternately exhibit high influences at different streamflow magnitudes (Fig. 6). Soil moisture controlled conditions (Ksconst) can be associated with the generation of fast reacting or slightly deferred streamflow components that dominate both at high and at low flows. At moderate high flows evapotranspiration (AspectcorrPET) intermittently dominates the streamflow simulations. The most sensitive parameters were additionally selected to highlight the parametric dominances along the independent variable of the FDCs. Hence, catchment-specific strips provided visually condensed diagnostic information for different streamflow

magnitudes.

TEDPAS analysis further clarified that the intermittent dominance of AspectcorrPET occurs simultaneously to the falling limbs subsequently to highflow peaks (Fig. 3c-f). In winter at higher altitudes this is compensated for instead by a proxy for snow dynamics (DegdayForest; Fig. 3b). Thus, the alternating sensitivity patterns of Ksconst and AspectcorrPET were jointly revealed by TEDPAS and INDPAS, both based on FDCs and on single-valued indices, where the pattern could be seperately

discerned for the Runoff Ratio (RR). Alternately to AspectcorrPET, again Ksconst showed significant sensitivities to the rest of the fingerprint metrics (Fig. 5). Overall, the combination of TEDPAS and INDPAS creates a clarified sensitivity pattern for different response characteristics of a catchment in the model space.

Regardless of the chosen model response target, in the case of 14 Ruhr headwaters only one or a very small group of parameters were identified as relevant for streamflow response. In this context, Herman et al. (2013) showed that the long-term

water balance is dominated by only very few parameters, irrespective of the hydrological conditions and of the model. Cuntz



et al. (2015) performed a global Sobol's sensitivity analysis on the hydrologic model mHM. For three distinct humid and arid European catchments they always resulted in about 20 informative parameters, though the dominant parameter sets were composed very differently. Their criteria to select the sensitive parameters was crucially different from our approach which renders a direct comparison between the studies difficult. The different number of dominant parameters might also be due to

correlated mHM parameters which we sorted out before sensitivity analysis. Differently, Cuntz et al. (2015) considered the degree of correlation between mHM parameters as rather minor to be disturbing for parameter identification.

The spread of the simulated response ensemble (e.g. FDCs) was found to differ between the catchments (section 3.3.2 and Fig. 6). This spread provides valuable information about the impact of the parameter variation on the simulation results. It can be indicative for the propensity of a catchment to certain hydrologic processes in the model space. This information is

important to judge whether a fingerprint metric is reliable as response target for sensitivity analysis or if different fingerprints, e.g. the master recession curve or the double mass curve, should be considered instead. In order to take into account the impact of the parameter variation on the spread of the simulation results, a weighting factor for partial parameter sensitivities might be helpful to select relevant parameters along with a meaningful response target for the specific hydrological conditions. The local impact of parameter variation on the model response might then be used along with catchment class-specific response

fingerprints within impact-weighted INDPAS to establish relationships of individual (vectors of) parameters and functional attributes.

In Fig. 6 we normalised the FDCs by the maximum value of each time series. For the visual comparison of sites this is a necessary step, but it might lead to a different form of appearance, including the spread of the simulation ensemble. It has thus to be kept in mind that dimensions crucially matter if real fingerprint values are replaced by normalised quantities (Samaniego

et al., 2010a; He et al., 2011), if a sensitivity weighting factor shall be determined.

## 4.2 Regional differences in parameter sensitivity

Although the most sensitive parameters and the corresponding sensitivity patterns of streamflow response were found to be similar for the 14 investigated Ruhr headwater catchments, the analysis with TEDPAS and INDPAS revealed certain regional differences.

Especially the analysis of sensitivity duration curves derived from TEDPAS revealed regional differences of parameter sensitivity between headwaters. For half of the eight selected parameters we found regional differences in SDCs (section 3.2 and Fig. 4). The most sensitive parameters exhibited the largest spread of SDCs (e.g. Ksconst and AspectCorrPET; Fig. 4c and e), and their SDCs were systematically ordered according to the geographical location (southwest-northeast) and the physiographic setting (EXP, URB; Table 3). Some catchments deviate from the general pattern in SDCs for evapotranspiration

and interflow parameters (e.g. RUE; Fig. 4e and f). In these cases, the specific combination of special characteristics (degree of soil sealing, topographic gradients, special land cover) might lead to preferential processes in streamflow simulations. For the catchment of RUE, the slightest slope value among the headwaters in conjunction with the highest percentage of urban area (SLOPE, URB; Table 3) can serve as explanation for the deviation from the general pattern of SDCs for the two parameters.



For the Ruhr headwaters, SDCs thus provide a convenient means to identify regionally different sensitivity characteristics for each of the analysed parameters.

The results from the INDPAS analysis of parameter sensitivity to single-valued fingerprints also showed some differences between the headwaters. In particular, the patterns of the second and third ranked parameters important for certain single-
valued fingerprints were found to differ, although not with the same systematic ordering as for the SDCs. The patterns of the most sensitive parameters along streamflow exceedance probability (catchment-specific strips in Fig. 6) in this case showed only minor differences between the catchments, but provide visually condensed diagnostic information for different streamflow magnitudes.

Together this shows that even the small physiographic gradients in the Ruhr headwater catchments can cause differences in
parameter sensitivity to streamflow response characteristics. This finding is partly contrary to those of Guse et al. (2014), who, in a similar analysis, reported almost no differences of parameter sensitivities among different subcatchments of the Treene in northern Germany. This was explained by the absence of a pronounced heterogeneity in their study area.

Given that the same parameter sets are applied to all headwater catchments, any regional differences in parameter sensitivity originate from differences either in the hydroclimatic or in the physiographic setting. In the case of the Ruhr headwaters,
the local hydroclimatic and physiographic differences (Table 3) seem to be sufficient to be discriminated by the hydrological model structure in the form of a different variation in streamflow response. Due to their geographical proximity, the 14 Ruhr headwaters are generally similar in terms of the longterm hydroclimate (Wetness Index; Table 3) and the physiography, e.g. in their soil hydrological characteristics (BFI; Table 3). They show, however, some differences in annual precipitation amount and in physiographic characteristics such as the topographic gradient and land cover (Table 3). The average annual precipitation in
the simulation period between 1997 and 2006 reveals a hydroclimatic gradient with lower to higher precipitation rates from northeast to southwest (PMEAN; Table 3) similar to the geographical ordering of the SDCs in Fig. 4. Likewise, Song et al. (2013) attributed local differences of parameter sensitivity to the spatial distribution of meteorologic forcing; Demaria et al. (2007) similarly concluded that parameter sensitivity was more strongly determined by climate gradients than by changes in soil properties in their Monte Carlo-based sensitivity study. Under different hydrological conditions regional sensitivity patterns
or the number of parameters which influence streamflow simulations might be different from the present example (Cuntz et al., 2015).

As parameter sensitivity is a prerequisite for parameter identifiability, even slight differences in sensitivity reveal information how identifiability can change among different catchments. Scale dependent limitations have to be kept in mind to avoid a levelling out of the explanatory value of a physiographic descriptor (Blöschl and Sivapalan, 1995), possibly resulting in
intermediate course of sensitivity duration as seen for the largest headwater of Bamenohl (BAM; section 3.2 and Fig. 4). Van Griensven et al. (2006) remarked that local differences indicate that results of global sensitivity analysis for one (group of) catchment cannot be directly applied to other, even closely located places, but may be used as reasonable estimates within the same catchment category. Places with intermediate sensitivity characteristics (e.g. BAM) could at least serve as starting point for parameter transfer to closely located and ungauged sites. As the local differences between the Ruhr headwaters are not
too large, the most sensitive parameters found for WEN in the first step of the analysis with all model parameters were also





dominant in the other subcatchments, which was corroborated by the TEDPAS analysis with eight selected parameters on all subcatchments. Any local differences in parameter sensitivity revealed by the analysis of sensitivity duration or INDPAS could then be handled during individual model calibration for each catchment.

## 5  Conclusions

We conducted a FAST global sensitivity analysis of the hydrological model mHM in 14 headwater catchments of the river Ruhr in western Germany. The two-tiered approach revealed that the simulation of streamflow response processes is most significantly influenced by soil moisture and evapotranspiration parameters. The parametric dominances along with the most influential parameters were found to change if one objective for sensitivity analysis is replaced, for instance, by another stream-flow response characteristic or when analysing along the range of streamflow magnitudes.

Catchment-specific patterns of parameter sensitivity that we unveiled by TEDPAS and INDPAS were found to slightly change even between relatively closely located places. These differences in sensitivity patterns are explainable not only by slight regional differences in the local hydroclimatic and topographic setting included in the model by the specific input data but also by their complex interplay controlling local hydrological functioning. However, the only slightly pronounced differences in the sensitivity patterns indicate that a parameter transfer to nearby catchments might be possible, provided that

the combination of catchment structure and local hydroclimate has not evolved crucially differently.

The methodical approach of the two-tiered sensitivity analysis may be generalised to any hydrological model or kind of catchment. The findings of this study motivate to include further catchments as regional end-members within different physio-graphic and climate settings. This is relevant to assess regional differences in the identifiability of parameters and to evaluate how parameter identifiability further changes among distant catchments. This step is also important to enable identification of

the regional relevance of model response targets for sensitivity analysis.

*Acknowledgements.* The streamflow data and the digital elevation model used in this study were provided by the Ruhrverband in Essen. We are grateful to four reviewers Fanny Sarrazin, Björn Guse, Shervan Gharari and one anonymous reviewer for their critical comments on an earlier version of the manuscript, which greatly helped to improve the paper. We thank Linda Bergmann who supported the technical implementation of the FAST analysis. We acknowledge support by Deutsche Forschungsgemeinschaft and Open Access Publishing Fund of

Karlsruhe Institute of Technology.





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



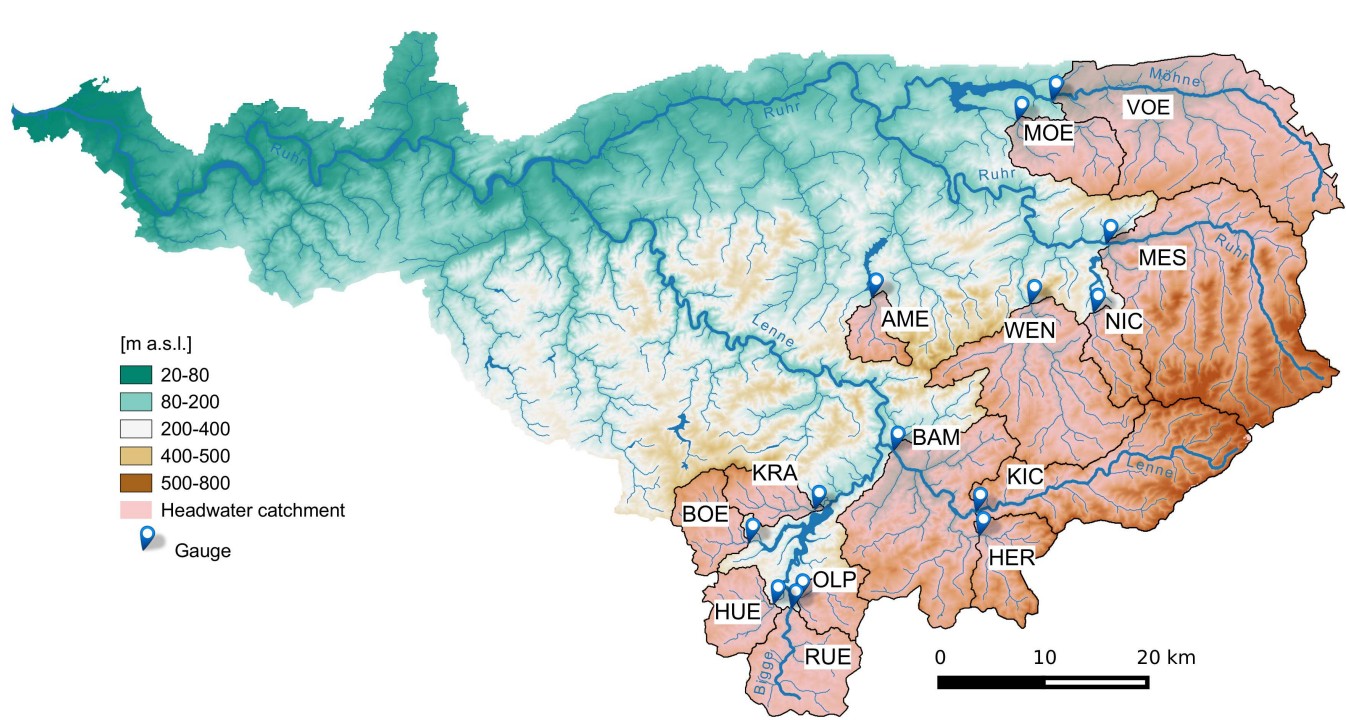

**Figure 1.** The Ruhr catchment with altitudinal zones, river network and 14 gauged headwater catchments: Amecke (AME), Bamenohl (BAM), Börlinghausen (BOE), Herrntrop (HER), Hüppcherhammer (HUE), Kickenbach (KIC), Kraghammer (KRA), Meschede1 (MES), Möhnesee-Neuhaus (MOE), Nichtinghausen (NIC), Olpe (OLP), Rüblinghausen (RUE), Völlinghausen (VOE) and Wenholthausen (WEN).



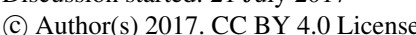

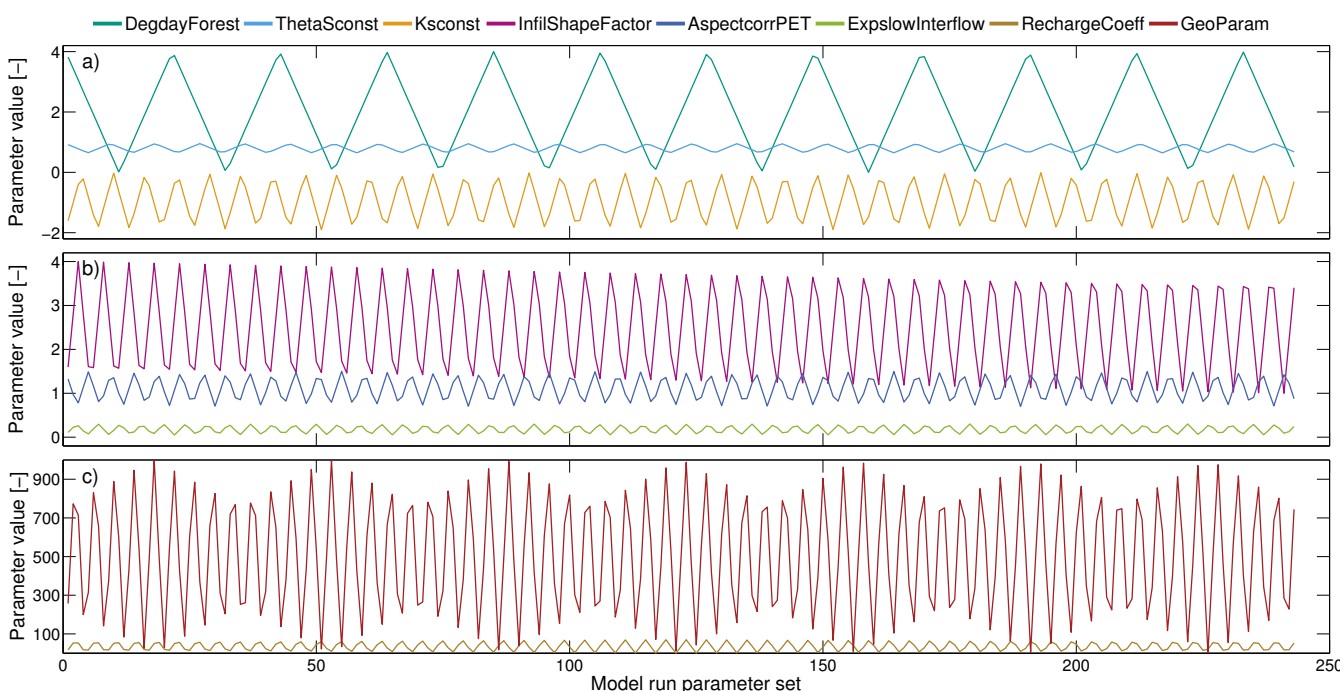

**Figure 2.** (a-c) Values of eight selected global mHM parameters for 243 model runs according to the FAST sampling plotted as connected curves (see also Table 2).





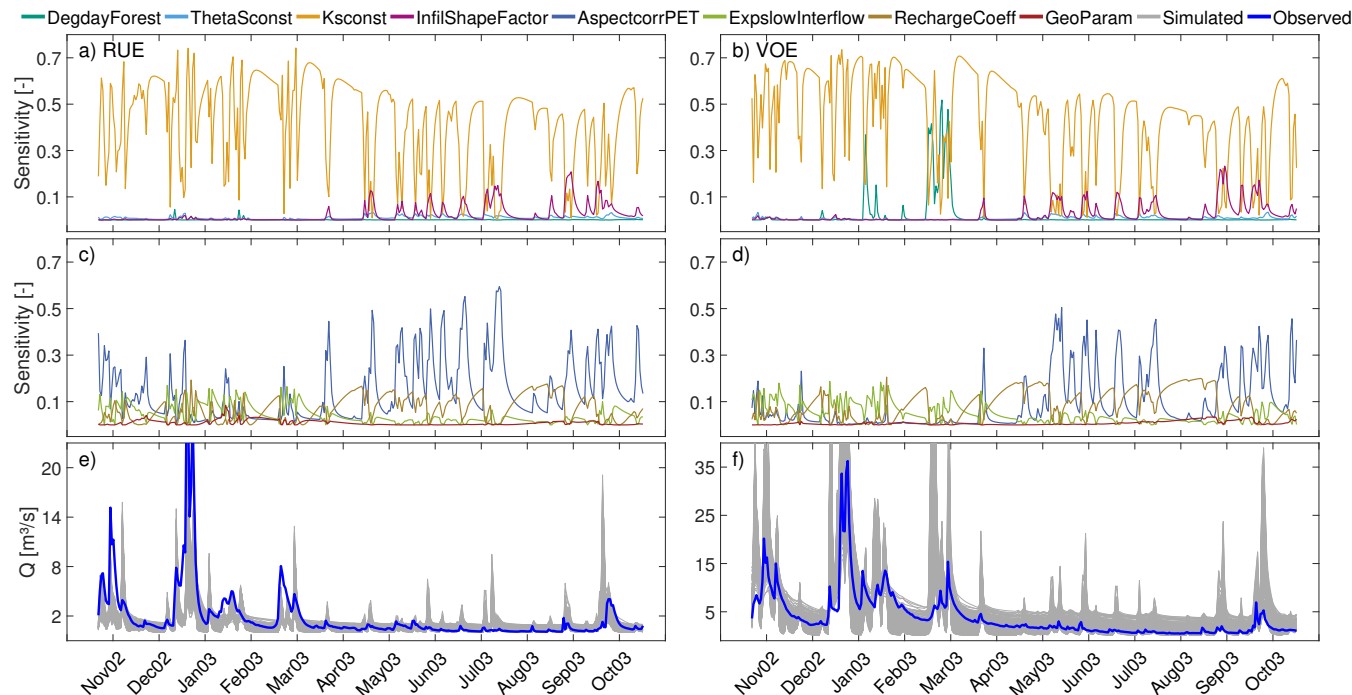

**Figure 3.** Time-dependent FAST sensitivities (TEDPAS) of eight global mHM parameters for two headwater gauges RUE (a, c) and VOE (b, d), and observed and FAST-mHM simulated streamflow ensembles at gauges RUE (e) and VOE (f). The results are shown for the hydrological year 2003 consisting of a wet season (November 2002 - March 2003) and a long dry period from April 2003 to September 2003. Please note the different axis scaling for streamflow for the two headwaters (e and f).





**Figure 4.** Sensitivity duration curves (SDCs) for eight global mHM parameters of 14 Ruhr headwater catchments (1997-2006): DegdayForest (a), ThetaSconst (b), Ksconst (c), InfilShapeFactor (d), AspectcorrPET (e), ExpslowInterflow (f), RechargeCoeff (g) and GeoParam (h). SDCs are shown normalised by the highest sensitivity value for each parameter among the headwaters. The eight parameters in terms of their average importance to streamflow simulations among all headwaters listed in descending order: Ksconst, AspectcorrPET, ExpslowInterflow, RechargeCoeff, ThetaSconst, InfilShapeFactor, GeoParam and DegdayForest.



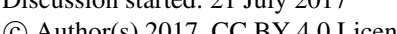


**Figure 5.** Parameter sensitivity for eight global mHM parameters to eight single value fingerprint metrics (RR, CV, HPC, SLFDC, HFD, BFI, RTC, ACT) in four Ruhr headwater catchments: RUE (a), VOE (b), HER (c), WEN (d). The simulation period was 1997 to 2006.




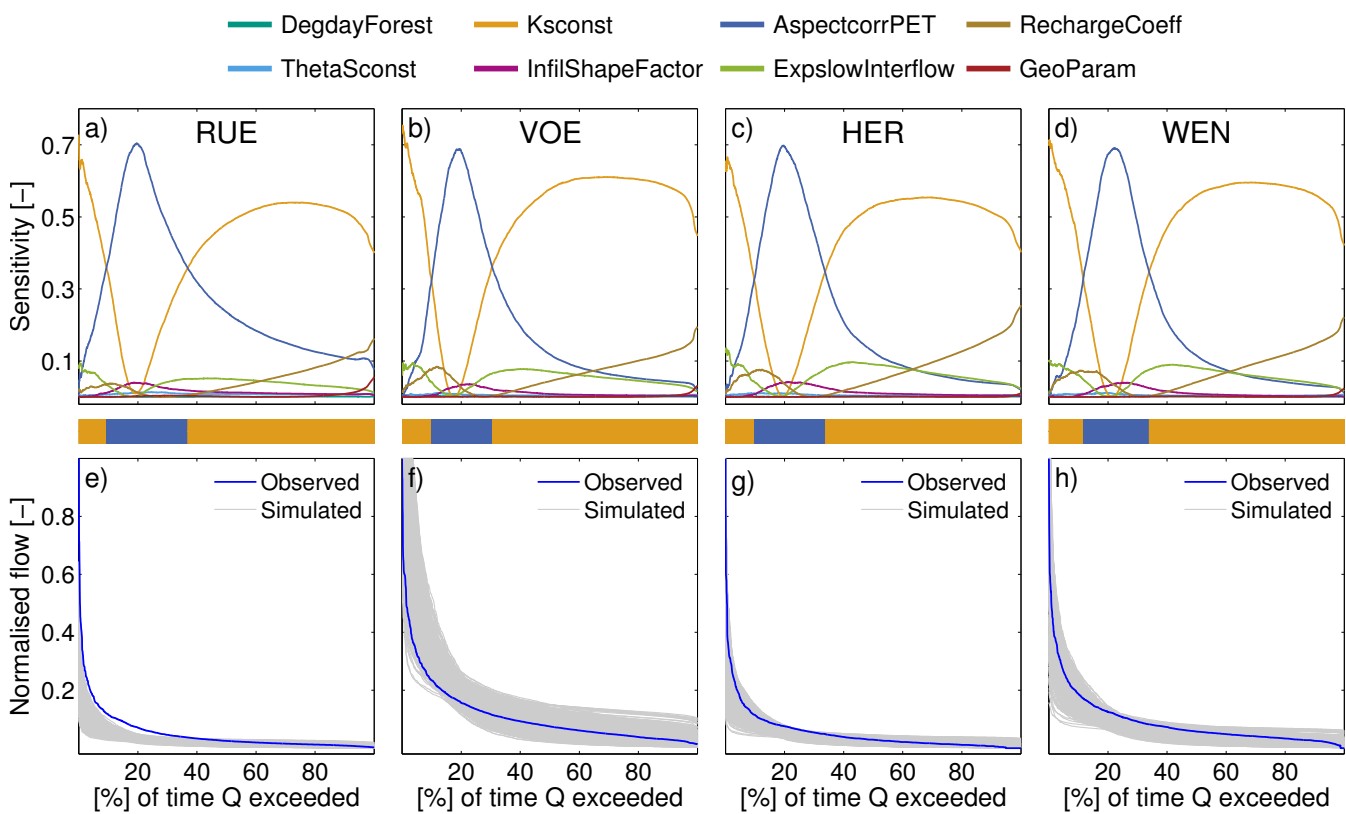

**Figure 6.** (a-d) Sensitivity for eight global mHM parameters along streamflow exceedance probability (flow duration curve, FDC) for four Ruhr headwater catchments RUE, VOE, HER and WEN. Catchment-specific strips show the parameters with the highest sensitivity along the FDC. (e-h) Observed flow duration curves and the corresponding ensembles of each 243 simulated FDCs for the period 2002 to 2006, normalised by the highest streamflow value observed at each of the four headwater gauges.





**Table 1.** Temporally aggregated single value fingerprint metrics derived from FAST-mHM simulated streamflow and observed precipitation time series, serving as model response targets for sensitivity analysis (INDPAS).

| Response Characteristic | Fingerprint metric | Abbreviation | [Unit] | Derivation |
|---|---|---|---|---|
| Water balance | Runoff Ratio | RR | [-] | $Q_{Total} / P_{Total}$ |
| Streamflow variability | Coefficient of Variation | CV | [-] | $\sigma / \mu$ |
| Frequency of flow events | High Pulse Count | HPC | [yr$^{-1}$] | (Number of timesteps $Q > 3 * Q_{mean}$) / years |
| Rate of change in streamflow | Slope of Flow Duration Curve | SLFDC | [%] | Slope of FDC between 33 % & 66 % Q exceedance |
| High flow conditions | High Flow Discharge | HFD | [-] | $Q_{5th\,percentile} / Q_{median}$ |
| Low flow conditions | Baseflow Index | BFI | [-] | $Q_{Baseflow} / Q_{Total}$ |
| Streamflow recession | Recession Time Constant | RTC | [d] | Mdn(Time required for $Q$ to reach $1/e * Q_{Peak}$) |
| Streamflow autocorrelation structure | Autocorrelation Time | ACT | [d] | Lag time required for AC function to decrease below 0.5 |

$Q$ = Streamflow; $P$ = Precipitation; $\sigma$ = Standard deviation; $\mu$ = Mean; Mdn = Median; AC = Autocorrelation



**Table 2.** Eight global mHM parameters (dimensionless): Function, value ranges, FAST sensitivity ranges and average sensitivity values from TEDPAS for 14 Ruhr headwater catchments (1997-2006).

| Parameter | Process | Description | Value Range [-] | Sensitivity Range [-] | Sensitivity Mean [-] |
|---|---|---|---|---|---|
| DegdayForest | snow | Determination of degree daily factor and maximum degree-day factor | 0 - 4 | 0 - 0.74 | 0.012 |
| ThetaSconst | soil moisture | Estimation of water content at saturation of soil (Constant part) | 0.65 - 0.95 | 0 - 0.68 | 0.018 |
| Ksconst | soil moisture | Estimation of saturated vertical hydraulic conductivity | -1.9 - 0.0 | 0 - 0.77 | 0.392 |
| InfilShapeFactor | soil moisture | Determination of numerical index of rooting distribution | 1 - 4 | 0 - 0.29 | 0.017 |
| AspectcorrPET | meteo correction | Account for aspect dependent correction of PET | 0.70 - 1.50 | 0 - 0.78 | 0.138 |
| ExpslowInterflow | interflow | Determination of exponent for the interflow reservoir | 0.05 - 0.3 | 0 - 0.22 | 0.062 |
| RechargeCoeff | percolation | Determination of percolation coefficient | 0 - 70 | 0 - 0.38 | 0.057 |
| GeoParam | geology | Determination of baseflow recession parameter | 0 - 1000 | 0 - 0.47 | 0.012 |



**Table 3.** Physiographic and climate descriptors characterising the topographic and hydroclimatic (1997-2006) setting of 14 Ruhr headwater catchments.

| Characteristic | Descriptor | Abbreviation | [Unit] | Headwater | | | | | | | | | | | | | |
|---|---|---|---|---|---|---|---|---|---|---|---|---|---|---|---|---|---|
| | | | | NIC | VOE | MOE | AME | BOE | RUE | HUE | OLP | KRA | KIC | HER | BAM | WEN | MES |
| 1) Climate | Wetness Index | P/PET | [-] | 2.3 | 2.1 | 2.1 | 2.4 | 2.5 | 2.2 | 2.3 | 2.2 | 2.2 | 2.4 | 2.3 | 2.3 | 2.2 | 2.5 |
| | Annual precipitation | PMEAN | [mm] | 1170 | 1025 | 1062 | 1240 | 1425 | 1286 | 1334 | 1232 | 1263 | 1261 | 1278 | 1237 | 1156 | 1192 |
| 2) Landform | Area | A | [km$^2$] | 37 | 293 | 66 | 29 | 48 | 86 | 47 | 35 | 38 | 187 | 61 | 453 | 184 | 426 |
| | Longest drainage path | LDP | [km] | 13 | 44 | 16 | 10 | 13 | 19 | 9 | 10 | 13 | 36 | 13 | 52 | 21 | 39 |
| 3) Topography | Slope | SLOPE | [%] | 22.6 | 10.9 | 11.2 | 17.3 | 14.9 | 10.8 | 13.2 | 17.5 | 16.0 | 26.1 | 21.1 | 22.5 | 16.2 | 20.9 |
| | Weighted slope | $I_g$ | [%] | 1.7 | 0.5 | 1.1 | 2.1 | 0.9 | 0.5 | 0.8 | 1.2 | 0.9 | 0.7 | 1.6 | 0.5 | 0.6 | 0.7 |
| | Elevation range | ELR | [m] | 430 | 426 | 316 | 361 | 297 | 394 | 204 | 274 | 391 | 533 | 415 | 425 | 440 | 585 |
| | Exposure | EXP | [°] | 195 | 189 | 187 | 189 | 167 | 175 | 170 | 197 | 159 | 201 | 199 | 188 | 186 | 190 |
| | Flow accumulation | FACC | [km$^{-1}$] | 0.7 | 0.3 | 0.6 | 0.7 | 0.6 | 0.5 | 0.5 | 0.7 | 0.9 | 0.5 | 0.5 | 0.2 | 0.3 | 0.2 |
| 4) Land cover | Forest | FOR | [%] | 55 | 50 | 87 | 51 | 54 | 43 | 40 | 62 | 52 | 72 | 75 | 67 | 41 | 60 |
| | Urban area | URB | [%] | 0 | 7 | 1 | 3 | 2 | 13 | 5 | 7 | 4 | 4 | 3 | 6 | 3 | 5 |
| | Pasture | PAST | [%] | 21 | 13 | 2 | 20 | 8 | 26 | 48 | 6 | 1 | 15 | 6 | 10 | 16 | 10 |
| | Cropland | CROP | [%] | 24 | 29 | 10 | 26 | 36 | 18 | 8 | 25 | 44 | 9 | 17 | 17 | 40 | 24 |
| 5) Soil | Baseflow index | BFI | [-] | 0.39 | 0.34 | 0.33 | 0.38 | 0.37 | 0.38 | 0.39 | 0.39 | 0.42 | 0.35 | 0.36 | 0.34 | 0.34 | 0.31 |