# Peer review of "Regional analysis of parameter sensitivity for simulation of streamflow and hydrological fingerprints"

_Hydrology and Earth System Sciences, 2017_

## Referee Comment (RC1) · B. Guse (Referee) · 9 Aug 2017

Review of the manuscript "Regional analysis of parameter sensitivity for simulation of streamflow and hydrological fingerprints" by Höllering et al.

In this manuscript, hydrological fingerprints are introduced as target variable for a sensitivity analysis and compared with a classical approach using streamflow data for a temporally resolved sensitivity analysis. The joint benefit of both approaches is presented for several headwater catchments.

As a reviewer of the first submission of this study to Hydrol. Earth Syst. Sci., I highly

appreciate the large effort of the authors. The manuscript is now clearer in its objective and I have only minor comments before publication. In most of the cases, my comments are suggestions. I hope that my comments help the authors to improve the manuscript.

P.3, L. 12-18: Even though that it is mentioned here and also at several other places, the manuscript could benefit from mentioning your contribution to the different options in selecting target variables for a sensitivity analysis in hydrology. These are: (i) Modelled discharge, (ii) Different objective functions (Van Werkhoven et al., 2008, Yilmaz et al., 2008, Herman et al. 2013), (iii) Different hydrological variables (Massmann and Holzmann, 2015; Guse et al., 2016), (iv) Different hydrological fingerprints (this manuscript). The use of fingerprint metrics as target variables for a sensitivity analysis is the novel contribution of this manuscript in my opinion. A clear listing could help in distributing this concept in the hydrological community.

P.5., L.7: I recommend to add some references that these metrics are often used in hydrology, since the approach depend on a reasonable selection of the fingerprint metrics. Probably it is sufficient to use studies which are already included in the reference list.

P.5., L.30: In the study of Van Griensven, FAST is listed as a type of sensitivity analyses, but I think that no FAST application is shown in this paper. This recommended to focus the referencing on the paper with FAST results.

P.6, L.9: The referencing of seven paper for mHm seems to be a high. Are they all required to justify the selection of model parameters and their ranges?

P.8, L.28: You may add that 2002 was a wet year and 2003 a dry year to show that you have consider different hydrological situations in the model calibration.

P.11, L.1-8: These results show clearly that the runoff ratio is a fingerprint metrics complementary to most of the others, while Fig.5 show similar results for several fingerprint

metrics. The study might benefit from a discussion of redundant and complementary fingerprint metrics. Based on this, it is probably possible to give some recommendations about an appropriate selection of a set of fingerprint metrics as target variables in sensitivity analyses.

P.12, L.5-12: Could you explain (based on its role in the model structure) why a soil moisture parameter is relevant both for high and low flows (but not for mid flow)? What about surface runoff or groundwater flow related parameter?

P.12, L.5-12: If it helps, you might add that the result that evapotranspiration parameters are sensitive in mid flows coincides with former sensitivity studies with other hydrological models.

P.12, L.28: I would recommend to emphasise this statement even more. You could discuss how this result helps for model calibration in terms of parameter constraining, reduction of parameter space and appropriate target variables.

P.13, L.1-5: Is ACT maybe less appropriate as fingerprint metric in sensitivity analysis? Also, at this point, it might be interesting to give some comments on the suitability on different fingerprint metrics.

P.13, L.12-13: At this point, I like to remark that Ksconst control both high and low flows. Thus, it would be interesting to see how precisely this parameter can be identified. I expect also conflicting results in model calibration studies. Even though that this study is not related to model calibration, your study give insights that this conflict might occur in the calibration.

References: The latter "a" is shown in smaller letters in the author list such as in P. 18, L. 28; P.20, L. 35; P. 21, L. 3.

Fig. 2: The information content of this figure is relatively low. I still see a reason for including it in the manuscript. However, in the case that you have to shorten the manuscript, you may think about removing this figure.

[Figure]

Fig. 3: For a better visualization, I recommend to show three subplots of sensitivity time series. It is very difficult to distinguish four lines in one subplot. Moreover, the blue colours are very similar. Maybe you can include the legend in the subplot (and increase the y-axis to 0.8).

Fig. 4: I recommend to move the last sentence from the figure caption to the main text. An interpretation of the results belongs to the main text and not to the figure caption.

Fig. 5: This figure clearly shows that a precisely definet target variable helps to disentangle the relevance of model parameters such as in this case for AspectcorrPET. This is an important result which could be more emphasised. It is clearly shown that separate target variables are beneficial for sensitivity analyses. These could be as denoted above: Different objective functions, different hydrological variables or such as in your study different fingerprint metrics.

Fig. 6: It seems to be that due to the high differences in sensitivity it is enough to select 4-6 parameters to control the hydrological behaviour in your case. Is this a typical number of relevant model parameters using mHm (model-inherent) or a case-specific result?

Tab. 2: I recommend to add the meaning of "Sensitivity range" and "Sensitivity mean" to a table caption.

References:

Guse, B., M. Pfannerstill, A. Gafurov, N. Fohrer, and H. Gupta (2016): Demasking the integrated information of discharge: Advancing sensitivity analysis to consider different hydrological components and their rates of change. Water Resources Research 52, 8724-8743, doi: 10.1002/2016WR018894

Herman, J. D., P. M. Reed, and T. Wagener (2013): Time-varying sensitivity analysis clarifies the effects of watershed model formulation on model behavior, Water Resour. Res., 49, 1400–1414, doi:10.1002/wrcr.20124.

Massmann, C., and H. Holzmann (2015): Analysing the sub-processes of a conceptual rainfall-runoff model using information about the parameter sensitivity and variance, Environ. Model. Assess., 20, 41–53, doi:10.1007/s10666-014-9414-6.

van Werkhoven, K., T. Wagener, P. Reed, and Y. Tang (2008): Characterization of watershed model behavior across a hydroclimatic gradient, Water Resour. Res., 44, W01429, doi:10.1029/2007WR006271.

Yilmaz, K. K., H. Gupta, and T. Wagener (2008): A process-based diagnostic approach to model evaluation: Application to the NWS distributed hydrologic model, Water Resour. Res., 44, W09417, doi:10.1029/2007WR006716.

---

## Author Comment (AC1) · 18 Sep 2017

Response to B. Guse (Referee)

**Björn Guse (BG)**: In this manuscript, hydrological fingerprints are introduced as target variable for a sensitivity analysis and compared with a classical approach using streamflow data for a temporally resolved sensitivity analysis. The joint benefit of both approaches is presented for several headwater catchments. As a reviewer of the first submission of this study to Hydrol. Earth Syst. Sci., I highly appreciate the large effort of the authors. The manuscript is now clearer in its objective and I have only minor comments before publication. In most of the cases, my comments are suggestions. I hope that my comments help the authors to improve the manuscript.

**Simon Höllering (SH): We sincerely thank Björn Guse for reviewing also our revised manuscript. We are grateful for the thoughtful recommendations and comments which will help to further clarify and improve the presentation of the study, especially for the hints on how to emphasize the main contributions of the present study. We will gladly follow his advice when revising the manuscript. Detailed comments on the specific points are collected below.**

**BG**: P.3, L. 12-18: Even though that it is mentioned here and also at several other places, the manuscript could benefit from mentioning your contribution to the different options in selecting target variables for a sensitivity analysis in hydrology. These are: (i) Modelled discharge, (ii) Different objective functions (Van Werkhoven et al., 2008, Yilmaz et al., 2008, Herman et al. 2013), (iii) Different hydrological variables (Massmann and Holzmann, 2015; Guse et al., 2016), (iv) Different hydrological fingerprints (this manuscript). The use of fingerprint metrics as target variables for a sensitivity analysis is the novel contribution of this manuscript in my opinion. A clear listing could help in distributing this concept in the hydrological community.

**SH: Thank you for this helpful comment. We will try to rework this part of the introduction to clarify our contribution.**

**BG**: P.5., L.7: I recommend to add some references that these metrics are often used in hydrology, since the approach depends on a reasonable selection of the fingerprint metrics. Probably it is sufficient to use studies which are already included in the reference list.

**SH: Yes, good point. We will add here some references.**

**BG**: P.5., L.30: In the study of Van Griensven, FAST is listed as a type of sensitivity analyses, but I think that no FAST application is shown in this paper. This recommended to focus the referencing on the paper with FAST results.

**SH: Yes, some of the studies present only reviews, others FAST applications and results. We will modify this part to become clearer here.**

**BG**: P.6, L.9: The referencing of seven paper for mHM seems to be a high. Are they all required to justify the selection of model parameters and their ranges?

**SH: Indeed, we will select the most relevant publications at this point.**

**BG**: P.8, L.28: You may add that 2002 was a wet year and 2003 a dry year to show that you have consider different hydrological situations in the model calibration.

**SH: We added this info in the caption of Fig. 3 to explain why these periods were selected for illustration. We will see how to incorporate this in the results or discussion parts dealing with this Figure and its interpretation. In section 2.4 we will also add that the period for initial model calibration was 1997-2006.**

**BG**: P.11, L.1-8: These results show clearly that the runoff ratio is a fingerprint metrics complementary to most of the others, while Fig.5 show similar results for several fingerprint metrics. The study might benefit from a discussion of redundant and complementary fingerprint metrics. Based on this, it is probably possible to give some recommendations about an appropriate selection of a set of fingerprint metrics as target variables in sensitivity analyses.

**SH: This is a good point that might be relevant to revise/add in the discussion. We are aware that redundancy in fingerprint metrics for a specific streamflow characteristic (e.g. high flows) might be possible if several similar metrics (e.g. HPC, CV, HFD) are selected. Nevertheless, a joint, multivariate analysis with metrics of several similar but slightly different streamflow characteristics (frequency of high flow, magnitude of high flows etc.) is needed to ensure complete parameter identification for different catchments. Our results also show that results in sensitivity analysis (INDPAS) can be similar even for fingerprint metrics that characterize different streamflow characteristics (compare RTC with CV, Fig. 5), which might change with the location.**

**BG**: P.12, L.5-12: Could you explain (based on its role in the model structure) why a soil moisture parameter is relevant both for high and low flows (but not for mid flow)? What about surface runoff or groundwater flow related parameter?

**SH: The widely largest importance of the soil moisture parameter is perhaps only masked and damped during mid flows by evapotranspiration parameters. The system state probably makes a large difference during high to mid flow conditions, because of the impact of evapotranspiration on the shallow soil storage in the catchment. This is especially the case for the recession limbs after peak flow, as we also specified in the manuscript: 'TEDPAS analysis further clarified that the intermittent dominance of AspectcorrPET occurs simultaneously to the falling limbs subsequently to high flow peaks (Fig. 3c and d, g and h).**
**The tested parameters for surface runoff (InfilShapeFactor) and groundwater (RechargeCoeff, GeoParam) were found to be much less sensitive, although the sensitivity of the latter increases towards the low flow conditions due to their impact on baseflow (Fig. 6), which seems plausible.**
**Our results provide in-depth diagnostics on the model, which can hopefully support future improvements, but an in-depth analysis of the mHM model structure is beyond the scope of the paper. We will try to include this in the discussion.**

**BG**: P.12, L.5-12: If it helps, you might add that the result that evapotranspiration parameters are sensitive in mid flows coincides with former sensitivity studies with other hydrological models.

**SH: Yes, we will check and possibly add studies.**

**BG**: P.12, L.28: I would recommend to emphasize this statement even more. You could discuss how this result helps for model calibration in terms of parameter constraining, reduction of parameter space and appropriate target variables.

**SH: Thank you, we will add some more explanation.**

**BG**: P.13, L.1-5: Is ACT maybe less appropriate as fingerprint metric in sensitivity analysis? Also, at this point, it might be interesting to give some comments on the suitability on different fingerprint metrics.

**SH: We will add here that regardless of the model parameterisation, precipitation has a large impact on the autocorrelation structure of streamflow. Thus, ACT is less informative than other metrics that depend less on the hydroclimatic boundary conditions.**

**BG**: P.13, L.12-13: At this point, I like to remark that Ksconst control both high and low flows. Thus, it would be interesting to see how precisely this parameter can be identified. I expect also conflicting results in model calibration studies. Even though that this study is not related to model calibration, your study give insights that this conflict might occur in the calibration.

**SH: Interesting remark. This would have to be tested by calibrating specifically for high and for low flow conditions. If large discrepancies in parameter values show up, an improvement of the model structure might be advisable.**

**BG**: References: The latter "a" is shown in smaller letters in the author list such as in P. 18, L. 28; P.20, L. 35; P. 21, L. 3.

**SH: Thank you for this hint. We have corrected it.**

**BG**: Fig. 2: The information content of this figure is relatively low. I still see a reason for including it in the manuscript. However, in the case that you have to shorten the manuscript, you may think about removing this figure.

**SH: We agree with Björn Guse and would prefer to include it in the paper, because it helps to understand the essence of the FAST method.**

**BG**: Fig. 3: For a better visualization, I recommend to show three subplots of sensitivity time series. It is very difficult to distinguish four lines in one subplot. Moreover, the blue colours are very similar. Maybe you can include the legend in the subplot (and increase the y-axis to 0.8).

**SH: Thanks for this recommendation. We reworked Fig. 3 now showing three subplots of sensitivity time series including the corresponding legends in each subplot with the proposed y-axes scaling. Additionally, for this figure, as well as for Fig. 2 and 6, we modified the shade of blue for one parameter (ThetaSconst). We think by these modifications the visualization has become clearer.**

**BG**: Fig. 4: I recommend to move the last sentence from the figure caption to the main text. An interpretation of the results belongs to the main text and not to the figure caption.

**SH: We will add the information in the text. However, we consider the information on the order of importance of the parameters as helpful to interpret the figure, especially the different shapes of the curves, and would thus leave the sentence here as well.**

**BG**: Fig. 5: This figure clearly shows that a precisely defined target variable helps to disentangle the relevance of model parameters such as in this case for AspectcorrPET. This is an important result which could be more emphasised. It is clearly shown that separate target variables are beneficial for sensitivity analyses. These could be as denoted above: Different objective functions, different hydrological variables or such as in your study different fingerprint metrics.

**SH: We agree with the reviewer that this is an important result and are happy to go with the recommendation to emphasize this a bit more in the text.**

**BG**: Fig. 6: It seems to be that due to the high differences in sensitivity it is enough to select 4-6 parameters to control the hydrological behaviour in your case. Is this a typical number of relevant model parameters using mHm (model-inherent) or a case-specific result?

**SH: Hard to say. It could be specific for the type of catchment, and/or the FAST approach. Other studies (cited in the paper P.14 L.1-6) found a larger number of parameters. For our FAST studies on the Ruhr headwaters it seems a typical number of relevant parameters (also in the preliminary and earlier studies).**

**BG**: Tab. 2: I recommend to add the meaning of "Sensitivity range" and "Sensitivity mean" to a table caption.

**SH: We have clarified the Table caption.**

**BG**: References:
Guse, B., M. Pfannerstill, A. Gafurov, N. Fohrer, and H. Gupta (2016): Demasking the integrated information of discharge: Advancing sensitivity analysis to consider different hydrological components and their rates of change. Water Resources Research 52, 8724-8743, doi: 10.1002/2016WR018894
Herman, J. D., P. M. Reed, and T. Wagener (2013): Time-varying sensitivity analysis clarifies the effects of watershed model formulation on model behavior, Water Resour. Res., 49, 1400–1414, doi:10.1002/wrcr.20124.
Massmann, C., and H. Holzmann (2015): Analysing the sub-processes of a conceptual rainfall-runoff model using information about the parameter sensitivity and variance, Environ. Model. Assess., 20, 41–53, doi:10.1007/s10666-014-9414-6.
van Werkhoven, K., T. Wagener, P. Reed, and Y. Tang (2008): Characterization of watershed model behavior across a hydroclimatic gradient, Water Resour. Res., 44, W01429, doi:10.1029/2007WR006271.
Yilmaz, K. K., H. Gupta, and T. Wagener (2008): A process-based diagnostic approach to model evaluation: Application to the NWS distributed hydrologic model, Water Resour. Res., 44, W09417, doi:10.1029/2007WR006716.

---

## Referee Comment (RC2) · S. Gharari (Referee) · 20 Sep 2017

The manuscript has improved significantly from the initial submission. However I am still struggling to get the main message of the paper. The title looks very broad. It is hard to predict what the reader should expect from the current title. The objectives and research questions are not answered in the conclusions. I still cannot easily summarize the manuscript scope and aim into one paragraph for myself! The organization of the paper is still far from being perfect. For example figure 1 is explained twice and in between there figure 2 is mentioned (I really do not get what the point of figure 2 is). The discussion part is too wordy and contains many repetition from the earlier sections

as well as literature review.

One more point which I am missing in the manuscript is, how the sensitivity was carried out on the fingering indices? Did the authors used the value of these indices and see how sensitive they are comparably? If this is the case how did the authors compare the sensitivities? the range of values for these indices might vary from one to the other.

I believe the manuscript should be better structured and the research questions should be better answered; for these reasons major revision is inevitable.

With regards

Shervan Gharari

---

## Author Comment (AC2) · 25 Sep 2017

Response to S. Gharari (Referee)

*Shervan Gharari (SG): The manuscript has improved significantly from the initial submission. However I am still struggling to get the main message of the paper.*

Simon Höllering et al. (SH): We sincerely thank Shervan Gharari for reviewing also our revised submission. We think we are able to overcome many of his concerns together with the revisions on the basis of the detailed comments of the first reviewer, Björn Guse. In summary, we will especially direct our efforts to a clearer presentation of the main contributions of the present study.

The detailed answers on the comments are collected below.

*SG: The title looks very broad. It is hard to predict what the reader should expect from the current title.*

SH: Thank you for your opinion and perception on the title. We have decided to choose a comparably brief title, which nevertheless points out the main features of our study. Admittedly, it does not summarize the contents of the paper in full detail. We would, however, prefer not to add more details to the title, as it probably would only be longer and not much clearer. Instead, we would invite the reader interested in "Regional sensitivity analysis" and "Hydrological fingerprints" to consult the abstract - and then hopefully the entire paper – for further information.

*SG: The objectives and research questions are not answered in the conclusions.*

**SH:** In this point we do not fully agree with the reviewer. We admit that the detailed answers to the research questions are contained in the discussion section, and that in the conclusions only short reference to the three questions is made.

Research question 1 ("Which parameters can be identified as sensitive with regard to specific hydrological response characteristics?") and research question 2 ("How does parameter sensitivity change with different hydrological objectives (response targets) applied in global sensitivity analysis?") are referred to in the first paragraph of the conclusions, while research question 3 ("How does parameter sensitivity change in different catchments under slightly distinct physiographic and hydroclimatic conditions?") is answered in the second paragraph.

We regard this structure to address the research question in the discussion and to summarize the main findings in the conclusion section as instructive and in line with many manuscripts, but would nevertheless happily consider any propositions on improvements of this issue.

*SG: I still cannot easily summarize the manuscript scope and aim into one paragraph for myself!*

**SH:** We will revisit the abstract and the short summary of our study once more to further optimize it for the reader. In this respect, reviewer 1 came up with a concise description that we will incorporate into the short summary: "In this manuscript, hydrological fingerprints are introduced as target

variable for a sensitivity analysis and compared with a classical approach using streamflow data for a temporally resolved sensitivity analysis. The joint benefit of both approaches is presented for several headwater catchments."

In addition to that, we will take care to point out more clearly the contribution of the study in our revisions of the introduction, discussion and conclusions as pointed out in our answers to reviewer 1.

*SG: The organization of the paper is still far from being perfect. For example figure 1 is explained twice and in between there figure 2 is mentioned (I really do not get what the point of figure 2 is).*

SH: Thank you for this comment. We are not sure if this is a major point; however, we will check the references to Fig. 1 (map of the study area) and delete any references which should not be needed to guide the reader to this map.

Figure 2 illustrates the variation of parameter values with different, independent frequencies along the number of simulation runs, which is at the heart of the FAST method. In our case, 243 (x-axis) different combinations of values for the eight parameters were obtained. Each parameter was varied in its specified ranges (y-axes). The parameter variations are displayed in three subplots for reasons of legibility.

243 model runs were performed and the same number of simulated streamflow time series was obtained. These 243 hydrographs are then used for TEDPAS and for the calculation of the fingerprint metrics for INDPAS.

We will add more explanation to the caption of Fig. 2 for clarification. But we still think that the figure is helpful and necessary, particularly for readers who are not familiar with FAST, to understand the essence of the method. We thus prefer to keep Fig. 2 in the revised manuscript.

*SG: The discussion part is too wordy and contains many repetition from the earlier sections as well as literature review.*

SH: Thank you for the comment. We will double check and optionally streamline the discussion.

For reasons of clarity, we structured the discussion in two parts. The first part addressed the complementary merits of TEDPAS and INDPAS. The second part focuses on the regional differences in parameter sensitivity among the headwaters and possible reasons for this, which are the specific findings for this hydrological system. While the latter might change when moving to a different study area, the former part is generic to the application of the proposed framework. Though one might choose other fingerprints in a different study, this does not change the philosophy of the approach. We will better stress why we structure the discussion that way in the revised manuscript.

*SG: One more point which I am missing in the manuscript is, how the sensitivity was carried out on the fingering indices? Did the authors used the value of these indices and see how sensitive they are comparably? If this is the case how did the authors compare the sensitivities? The range of values for these indices might vary from one to the other.*

SH: Maybe the reviewer missed the description of how the sensitivity analyses were carried out (Sections 2.2 and 2.5.2)? If anything should be unclear about these paragraphs, we would appreciate any hints.

The sensitivity analyses were performed with the Fourier Amplitude Sensitivity Test (FAST, section 2.2 in the manuscript). The general idea of FAST is to vary parameters of interest with independent frequencies, and then perform a Fourier analysis of the simulated target variable across the ensemble of model runs to determine the first-order sensitivities of the parameters as a function of time, hence obtaining a power spectrum for each simulation time step. The variance $\sigma_i^2$ that is explained by a parameter *i* is determined by normalizing the corresponding power with the total power in the spectrum, which corresponds to the total variance $\sigma^2$ within the model ensemble (see the cited paper by Reusser et al. 2011). The sensitivity of model output on parameter *i* is then calculated as the partial variance, which is the ratio $\sigma_i^2/\sigma^2$.

We tested different model results (hydrograph and fingerprint metrics, i.e. single value indices and FDCs) in our analyses, which yielded parameter sensitivities with respect to each of these target variables. The fingerprint metrics were calculated from each of the 243 simulated streamflow hydrographs, and each one was individually analyzed with FAST. This analysis directed to fingerprint metrics is essentially the same procedure as for TEDPAS. The resulting partial variance for each fingerprint are comparable because they portray the relative influence of the parameters on the variation of the target, regardless of the concrete values of the targets, which might indeed be quite different.

The range of model results, for example in single value fingerprints, might influence the sensitivity analyses as discussed in section 4.1 with the example of the autocorrelation time (ACT). Following the recommendation of the first reviewer, we will add some more explanation in the discussion on this topic. We could also try to further clarify the explanation of the INDPAS method, if not already sufficiently explained.